# Biomarkers of Response and Resistance to Immunotherapy in Microsatellite Stable Colorectal Cancer: Toward a New Personalized Medicine

**DOI:** 10.3390/cancers14092241

**Published:** 2022-04-29

**Authors:** Nicolas Huyghe, Elena Benidovskaya, Philippe Stevens, Marc Van den Eynde

**Affiliations:** 1Institut de Recherche Clinique et Expérimentale (Pole MIRO), UCLouvain, 1200 Brussels, Belgium; nicolas.huyghe@uclouvain.be (N.H.); elena.benidovskaya@student.uclouvain.be (E.B.); philippe.stevens@uclouvain.be (P.S.); 2Institut Roi Albert II, Department of Medical Oncology and Gastroenterology, Cliniques Universitaires St-Luc, 1200 Brussels, Belgium

**Keywords:** colorectal cancer, immunotherapy, Immune Checkpoint Inhibitors, immune checkpoint resistance, immune microenvironment, biomarker

## Abstract

**Simple Summary:**

Immune Checkpoint Inhibitors (ICIs) have demonstrated clinical efficacy in Microsatellite Instability High Colorectal Cancer (MSI-H CRC). However, in Microsatellite Stable (MSS) CRC, ICIs monotherapy provides limited clinical benefit. Therefore, efforts must be made to understand the highly heterogeneous CRC microenvironment and to find predictive biomarkers of response in order to adequately select CRC patients who may respond to ICIs-based therapies.

**Abstract:**

Immune Checkpoint Inhibitors (ICIs) are well recognized as a major immune treatment modality for multiple types of solid cancers. However, for colorectal cancer (CRC), ICIs are only approved for the treatment of Mismatch-Repair-Deficient and Microsatellite Instability-High (dMMR/MSI-H) tumors. For the vast majority of CRC, that are not dMMR/MSI-H, ICIs alone provide limited to no clinical benefit. This discrepancy of response between CRC and other solid cancers suggests that CRC may be inherently resistant to ICIs alone. In translational research, efforts are underway to thoroughly characterize the immune microenvironment of CRC to better understand the mechanisms behind this resistance and to find new biomarkers of response. In the clinic, trials are being set up to study biomarkers along with treatments targeting newly discovered immune checkpoint molecules or treatments combining ICIs with other existing therapies to improve response in MSS CRC. In this review, we will focus on the characteristics of response and resistance to ICIs in CRC, and discuss promising biomarkers studied in recent clinical trials combining ICIs with other therapies.

## 1. Introduction

Although there has been recent progress in management, treatment, and screening, Colorectal Cancer (CRC) remains a major public health issue. Worldwide, CRC is estimated to be the third most frequent cancer and the second cause of cancer-related death [1]. Despite a major emphasis on CRC screening, approximately 20% of CRC is metastatic at diagnosis and 30% of treated non-metastatic patients will develop metastasis during the follow-up of their disease [2]. Metastatic CRC (mCRC) has a poor prognosis with a 5-year survival rate of 14.2% (95% CI, 13.7–14.7) [3]. Treatment options for CRC include surgery, chemotherapy, radiation, targeted therapy, and, more recently, immunotherapy for a selected molecular subgroup of tumor [4].

Based on genetic alterations such as mutations in V-Ki-ras2 Kirsten Rat Sarcoma viral oncogene homolog (*KRAS*) and proto-oncogene B-Raf (*BRAF*), and mutations or methylation of Mismatch Repair (MMR) genes, several specific treatments of molecular CRC subgroups have emerged over recent years [5]. Immune Checkpoint Inhibitors (ICIs) demonstrated good efficacy in deficient-Mismatch-Repair Microsatellite Instability-High (dMMR/MSI-H) CRC, providing clinical benefits superior to standard treatments and leading to the Food and Drug Administration (FDA) and European Medicines Agency (EMA) approval of anti-Programmed cell Death protein 1 (anti-PD-1) ICIs for the treatment of metastatic or unresectable dMMR/MSI-H CRC [6,7,8]. Nonetheless, for the vast majority of Microsatellite Stable (MSS) CRC, ICIs failed to provide clinical benefit in unselected cohorts [9]. Therefore, it is crucial to better understand the genetic, epigenetic, transcriptomic, and Tumor Microenvironment (TME) characteristics of the MSS CRC.

In this review, we will discuss the genomic, transcriptomic, and tumor microenvironment classifications of CRC in an immunogenic way. This will help in understanding the different biomarkers of response and resistance to ICIs investigated so far as well as promising clinical trials recently developed combining these biomarkers with therapies to improve the adaptive immune response of the tumor and the benefit of ICIs to the patient.

## 2. Colorectal Cancer Subtypes and Immunity

Several classifications have been performed and are now available to understand CRC development and progression, estimate prognosis, and select patients potentially able to respond to specific treatments. In this section, we discuss genomic, epigenetic, transcriptomic, and TME alterations and their relationship to CRC immunity.

### 2.1. Genomic and Epigenomic Classifications

At the genomic level, it is well recognized that the majority of CRC (85%) presents a Chromosomal Instability (CIN) and is MSS while a minority of CRC (15% in early stages) is characterized by an MSI-H phenotype (Figure 1) [10]. The microsatellite instability occurs through a deficiency in the mismatch repair mechanism that corrects single nucleotide base mispairings and small insertions or deletions (indels) appearing during DNA replication [11], leading to the accumulation of such alterations across the genome, favoring the development of cancer [12]. This loss of function of any of the DNA mismatch machinery proteins, including MutS Homolog 2 (MSH2), MutS Homolog 6 (MSH6), MutL Homolog 1 (MLH1), and Postmeiotic Segregation Increased 2 (PMS2) [13] leads to a “hypermutated” phenotype. These tumors are characterized by high Tumor Mutational Burden (TMB) leading to a high number of Mutation-Associated Neoantigens (MANA) and by consequence an inflammatory TME, comprising Tumor-Infiltrating Lymphocytes (TILs), and notably memory cells and Cytotoxic T Lymphocytes (CTLs). In 12% of the cases, the deficiency is sporadic and occurs through the silencing of the MLH1 gene by promoter hypermethylation. In 3% of the cases, the deficiency occurs through germline gene mutation (Lynch syndrome) and, in rare cases, through somatic bi-allelic mutations [10,14]. In a systematic review including 1277 MSI-H out of 7642 stage I–III cases, it has been shown that the pooled Hazard Ratio (HR) for Overall Survival (OS) was 0.65 (95% CI, 0.59–0.71) for MSI-H versus MSS, suggesting a better prognosis for MSI-H patients [15].

The DNA Polymerase Exonuclease Domain (*POLE/POLD1*) mutations could also induce a high MANA load and immune infiltration (Figure 1). These mutations can affect the proofreading function of the polymerase, resulting in an “ultramutator” phenotype and patients harboring *POLE/POLD1* mutations are prone to developing CRC [16,17]. The frequency of *POLE/POLD1* mutations in CRC is near 1%, explaining the lack of robust clinical and translational data on this subgroup of CRC [18,19]. However, in a recent report, it has been shown that similarly to MSI-H tumors, *POLE/POLD1*-mutated tumors correlate with a higher density of cytotoxic T cells (CD8+) and memory T cells (CD45RO+) compared to *POLE/POLD1* wild-type tumors [20].

Mutation in the *KRAS* driver oncogene has recently been shown to impact immunomodulation [21]. *KRAS*-mutant CRC patients have more T regulatory lymphocytes (Tregs) and fewer activated CD4+ memory T cells as compared to *KRAS* wild-type patients, resulting in a more immunosuppressive TME [22]. The loss of Adenomatous Polyposis Coli (*APC*) tumor-suppressor gene, involved in the cellular transition from G1 to S phase, can result in the sustained activation of the proto-oncogene *Wnt* (*Wnt*) signaling pathway and, thus, increased nuclear β-catenin [23] and decreased T cell tumor infiltration [24]. Activating mutations in Phosphatidylinositol-4,5-Bisphosphate 3-Kinase Catalytic Subunit Alpha (*PIK3CA),* occurring in 25% of CRC cases [25], are associated with CD8+ T cell infiltration and Programmed Death-Ligand 1 (PD-L1) expression [26]. CRC studies also reported that a loss of Phosphatase and TENsin homolog *(PTEN),* regulating PI3K/AKT signaling pathway, results in increased PD-L1 expression [27], decreased TILs presence, and an immunosuppressive TME [28].

Epigenetic regulation is an important hallmark of cancer cells [29]. In CRC, epigenetic alterations may account for the distorted expression patterns of certain genes without genetic alteration through hypermethylation or hypomethylation. Hypermethylation occurs when methyl groups are covalently bound in regions of CG dinucleotides or CpG-rich areas of DNA in promoter regions. In instances of normal gene expression, CpG regions are normally maintained in the unmethylated state. Methylation of these promoter regions can lead to gene silencing in tumor-suppressor genes, which is denoted as CpG Island Methylator Phenotype (CIMP). The CIMP has been defined and associated with the MSI-H phenotype, older age at diagnosis, right-sided location, mucinous histology, *BRAF* mutation, and a high T cell infiltration (Figure 1) [30]. Nevertheless, its use is limited by the absence of consensus regarding which genes should be included in the CIMP panels [31]. Global DNA hypomethylation can also lead to tumorigenesis and chromosomal instability in CRC [32]. Various preclinical studies investigate treatments targeting epigenetic changes since these latter are reversible [33,34]. Nevertheless, few clinical studies consider the potential synergic effects of methylation inhibitors. A phase I/II clinical trial studying the association of 5-azacitidine (DNA methyltransferase (DNMT) inhibitors) with capecitabine and oxaliplatin (CAPOX) in refractory CIMP-high mCRC demonstrated the safety and efficacy (high rate of disease control rate) of this association [35]. The use of DNMT inhibitors is limited by its lack of specificity targeting the global methylation of normal and oncogenic genes [36].

### 2.2. Transcriptomics Classification

Based on gene expression data from 18 public data sets and The Cancer Genome Atlas (TCGA) proprietary data sets, four subgroups of CRC have been identified by combining six different CRC subtyping algorithms forming the Consensus Molecular Subtypes (CMS) of CRC [16]. Briefly, CMS1 subtype (“immune subtype”, 14% of CRC tumors) is characterized by a high TMB and MANA load, high immune infiltration, T helper 1 (Th1) signaling, *BRAFV600E* mutations, and an overexpression of immune checkpoint molecules such as PD-1, Cytotoxic T-Lymphocyte-Associated protein 4 (CTLA-4), and Indoleamine 2,3-Dioxygenase 1 (IDO1). This subtype is mainly composed of MSI-H tumors, but it has been shown that around 16% of MSS tumors show characteristics of CMS1 such as high immune infiltration [37]. The CMS2 subtype (37% of CRC tumors) is called “canonical”, highlighting its epithelial features and activation of the WNT and Myelocytomatosis oncogene Myc (MYC) pathways. This group is enriched by CRC with the lowest MSI-H rate (less than 2%) [38]. Due to the low TMB, the infiltration of immune cells is very low, making them known as the “immune-desert” CRC subtype [16,39]. The CMS3 subtype (“metabolic subtype”, 13% of CRC tumors) is characterized by a perturbation of metabolic pathways and a high prevalence of *RAS* mutations. Immune cell infiltration in CMS3 is slightly higher than CMS2 but is still low and has an immunologically inactive TME, referred to as “immune-excluded” subtype (10). Eventually, CMS4 tumors (23% of CRC tumors) are called “mesenchymal type” because they have mesenchymal properties such as strong endothelial-mesenchymal transition activity and high stromal content with cancer-associated fibroblasts [16]. They are highly infiltrated with immunosuppressive cells such as Tregs, M2 macrophages, and Myeloid-Derived Suppressor Cells (MDSCs), and the presence of antitumor immune cells such as Dendritic Cells (DCs), activated Natural Killer (NK) cells, Th1 cells, and CD8+ T cells in their TME is very low. In addition, CMS4 tumors have activated Vascular Endothelial Growth Factor (VEGF), Transforming Growth Factor beta (TGF-β), and Chemokine (CXC motif) Ligand 12 (CXCL12) signaling pathways, all of which cause, among the four CMS categories, higher risk of relapse and worst prognosis of CMS4 CRC after surgery [16].

However, these four CMS subtypes have been identified using expression data from primary lesions and it is difficult to transpose this classification in mCRC [40]. In a recent study, authors applied the CMS classification on resected CRC Liver Metastasis (CRCLM) and reported that, conversely to primary CRC, the CMS classification in CRCLM was not associated with OS. Using transcriptional data of messenger RNA (mRNA) and micro RNA (miRNA), they derived three subtypes of CRCLM (subtype 1: canonical, subtype2: immune, subtype 3: stromal) with different clinical risk stratification for Disease-Free Survival (DFS) and OS [40]. Another study reported a similar depletion of CMS1 and CMS3 subtypes in 295 CRCLMs [41]. Due to tumor heterogeneity [42,43,44], they also observed frequent CMS switches between CRCLM of the same patients [41]. Together, these reports suggest that the CMS classification may not be suitable for mCRC.

### 2.3. Classification Regarding the Tumor Microenvironment

#### 2.3.1. The Tumor Immune Microenvironment

Besides molecular classification, CRCs can also be classified based on immunological properties. As mentioned, the mutational burden is a hallmark for the response to immunotherapy. However, the resistance of some MSI-H tumors and the appropriate response of some MSS tumors suggest that a high TMB is not responsible alone for the response to immunotherapy [45]. CRCs with high immune infiltration are not only composed of hypermutated tumors and CMS1 subtypes [46]. The CRC TME is a heterogeneous microenvironment containing a variety of immune cells. Numerous studies have linked high infiltration of CD8+ cytotoxic T cells, Th1, follicular helper T cells, M1 macrophages, NK cells, and DCs with good prognosis in CRC. Contrarily, high infiltration of MDSCs, B cells, M2 macrophages, and CD4+ type-17 helper T (Th17) cells is associated with poor prognosis [47,48,49].

The characterization of the immune infiltration became an important tool to classify the CRC tumor subtypes. In 2006, J. Galon et al. provided strong evidence that the type, density, location, and functional orientation of T cells infiltration was associated with favorable DFS and OS after primary CRC (stage I–III) resection [50,51,52,53]. They developed the Immunoscore, simple and reproducible scoring that could be used routinely to predict patient clinical outcome. The Immunoscore (I) is computed by using the density of T cells (CD3+) and cytotoxic (CD8+) T cells in the Tumor Center (CT) and the Invasive Margin (IM) of the tumor. The Immunoscore ranges from I0 to I4. Low density of both CD3+ and CD8+ T cells in the CT and the IM is associated with I0 while high density is associated with I4 [51,52,54].

In 2018, an international consortium of 14 centers in 13 countries, led by the Society for Immunotherapy of Cancer, assessed the Immunoscore assay in 2681 patients with TNM stage I–III colon cancer (CC). The Immunoscore assay showed a high level of reproducibility between observers and centers (r = 0·97 for colon tumor; r = 0.97 for invasive margin; *p* < 0·0001). Of 1434 patients with stage II cancer, the difference in risk of recurrence at 5 years was significant (HR for high vs. low Immunoscore: 0.33, 95% CI 0.21–0.52; *p* < 0·0001), including in the Cox multivariable analysis (*p* < 0·0001). Immunoscore had the highest relative contribution to the risk of all clinical parameters, including the American Joint Committee on Cancer and Union for International Cancer Control TNM classification system [55]. Similar results were found for stage III CC [55]. The IDEA-France prospective study [55], together with another phase 3 randomized clinical trial (N0147) [56], validated the value of Immunoscore in prognostication of relapse and death in stage III CC patients treated with adjuvant treatment combining fluoropyrimidine and oxaliplatin. The IDEA France clinical trial, evaluating 3 versus 6 months of oxaliplatin-based adjuvant chemotherapy, demonstrated the predictive value of Immunoscore for treatment duration. Immunoscore predicted response to 6 months folinic acid, fluorouracil, and oxaliplatin (FOLFOX) chemotherapy both in low- and high-risk stage III patients. Low-risk patients (T1-3, N1) with High-Immunoscore had 3-year DFS of 91.4% when treated with the 6-month FOLFOX, and only 80.8% with the 3-month regimen. These results and recent guidelines argue for the benefit of implementing the Immunoscore in clinical practice and for its introduction in a new TNM-Immune (TNM-I) classification system.

In mCRC, high immune and genetic heterogeneity between the different synchronous and metachronous metastases of the same patients has been reported [37,57]. High T cell infiltration and Immunoscore measured in the least-infiltrated metastasis were associated with a significantly lower number of metastases, larger metastasis, and prolonged survival while patients with increased metastatic burden generally had a lower Immunoscore [57].

#### 2.3.2. The Cancer-Associated Microbiome

It is becoming clear that microbes exist outside of the gastrointestinal tract, and the interplay of gut, circulating, and tissue-resident microbiomes with the development and treatment of malignancy is being explored. Among seven tumor types outside the gastrointestinal tract, Nejman et al. [58] assessed the microbiome of 1526 human cancers or adjacent normal tissue, taking multiple measures to avoid contamination and using 5R multiplexed bacterial 16S ribosomal DNA (rDNA) Polymerase Chain Reaction (PCR) sequencing. They found that each tumor type has a distinct microbiome composition. The intratumor bacteria are mostly intracellular and are present in both cancer and immune cells (CD45+ and CD68+ cells). Authors also reported correlations between intratumor bacteria and tumor types and subtypes and response to immunotherapy. These results are consistent with a recent publication which reexamined whole-genome and whole-transcriptome sequencing studies in the TCGA of 33 types of cancer for microbial reads [59]. Authors found unique microbial signatures in tissue and blood within and between most major types of cancer which remained predictive when applied to patients with non-metastatic or early cancer and cancers lacking any genomic alterations.

CRC patients with microbiotas enriched for pathogenic bacteria such as *Fusobacterium nucleatum* (*Fn*) and *Bacteroides fragilis* (*Bf*) generally present distinct phenotypic (frequent association with MSI-H, *BRAF* mutation, right-sided location) and clinical features (impaired chemotherapy response and poor outcome) [60]. One single paper [61] reported that *Fn*, a Gram-negative anaerobic bacterium, and other associated bacteria (*Bf*, *Selenomonas*, …) were, next to the primary tumor, also present inside cancer cells of CRC metastases. They observed that mouse xenografts of human primary CRC were found to retain viable bacteria including *Fn* through successive passages and that treatment with metronidazole reduced *Fn* load, cancer cell proliferation, and tumor growth. Experiments in vitro showed that Fn triggered innate immune signaling and induced specific genomic loss of miRNAs miR-18* and miR-4802 targeting Unc-51-Like Autophagy-Activating Kinase 1 (*ULK1*) and Autophagy-Related 7 (*ATG7*), respectively. Thus, *Fn* causes chemoresistance by selectively targeting specific miRNAs and autophagy pathways [62]. A recent study observed that *Fn* persistence in locally advanced rectal cancer after preoperative chemoradiotherapy was associated with higher risk of cancer relapse after surgery [63]. Interestingly, authors suggest a possible immunological mechanism for worse outcome. Tumors that turned *Fn*-negative after preoperative Chemoradiotherapy (CRT) had a strong increase in CD8+ T cells, while those that remained *Fn*-positive after treatment lacked CD8+ T cells induction as compared to baseline. This suggested that *Fn* may promote a lack of immune cytotoxicity activation [64] and may favor metastatic spread. The influence of the CRC microbiome and its relationship with anticancer immunity raises new questions from preclinical and clinical standpoints. However, besides these initial insights in microbiota-primary tumor interaction, the role of bacteria in CRC metastases remains obscure.

As shown in Figure 2 (adapted from [65]), all these classifications of CRC overlap strongly. Each represents a unique way of representing and subdividing the different subgroups of CRC.

## 3. Immune Checkpoint Inhibitors in Colorectal Cancer

Immune Checkpoint Inhibitors (ICIs) have been developed to block co-inhibitory signals that regulate the effector T cells response. These co-inhibitory signals are called “immune checkpoint” and ICIs are treatments, usually monoclonal antibodies, which block co-inhibitory signals and improve immune activity in the tumor and the blood of the patients by preventing the dysfunction and apoptosis of T effectors [66,67]. The most exploited therapeutic targets are PD-1, PD-L1, and CTLA-4, but there is a plethora of other co-inhibitory or co-stimulatory checkpoint molecules such as Lymphocyte-Activation Gene 3 (LAG-3), T cell Immunoglobulin and Mucin-containing protein-3 (TIM-3), and Tumor Necrosis Factor Receptor superfamily, member 4, also known as OX40 receptor, that can be targeted and are currently under investigation in several trials [45]. In this section, we summarize the current efficacy of ICIs (anti-PD-1-L1 treatment combined or not with anti-CTLA-4) in MSI-H and MSS CRC.

### 3.1. ICIs in MSI-H CRC

The first durable Complete Response (CR) observed with ICIs was observed in a phase I study evaluating nivolumab (anti-PD-1) in the treatment of refractory solid tumors. The CR lasted longer than 3 years and the patient had in fact MSI-H mCRC, underlying the potential of ICIs in this subset of CRC [68,69].

Following these initial findings, other trials (KEYNOTE-016, KEYNOTE-164, KEYNOTE-158, KEYNOTE-012, and KEYNOTE-28) evaluating pembrolizumab (anti-PD-1) for the treatment of refractory MSI-H mCRC have been conducted (Table 1). In total, 90 patients were evaluated. The Overall Response Rate (ORR) was 39.6% and lasted over 6 months in 78% of patients. These results led in 2017 to the fast FDA approval of pembrolizumab for MSI-H chemo refractory mCRC [7].

The non-randomized multicohort CheckMate 142 trial, evaluating the safety and efficacy of nivolumab 3 mg/kg combined or not with ipilimumab 1 mg/kg 4 doses (anti-CTLA-4) once every 3 weeks in chemo refractory mCRC, reported an ORR and 1-year OS rate of 55% and 85% for the treatment with nivolumab—ipilimumab compared to, respectively, 31% and 73.4% for nivolumab monotherapy [70,71]. Grade 3 to 4 treatment-related adverse events (AEs) occurred in 32% of patients treated with the combo compared to 21% with nivolumab monotherapy, and were manageable. This trial suggests that the combination of ipilimumab with nivolumab could be superior to nivolumab monotherapy for the treatment of MSI-H chemo refractory mCRC. The FDA also approved nivolumab, with or without combination with ipilimumab for the treatment of previously treated MSI-H CRC.

The randomized phase III KEYNOTE-177 trial evaluated the safety and efficacy of pembrolizumab in treatment-naïve MSI-H mCRC patients (47). Patients were treated either with pembrolizumab 200 mg every 3 weeks or with chemotherapy ± cetuximab or bevacizumab as Standard of Care (SOC). Treatment-related adverse events of grade 3 or higher occurred in 22% of the patients in the pembrolizumab group, as compared with 66% in the chemotherapy group. At final analysis, median overall survival was not reached (NR; 95% CI 49·2–NR) with pembrolizumab vs. 36·7 months (27·6–NR) with chemotherapy (HR 0·74; 95% CI 0·53–1·03; *p* = 0·036). The estimated median progression-free survival (PFS) was 16.5 months (95% CI: 5.4–32.4) versus 8.2 months (95% CI: 6.1–10.2) in the pembrolizumab and SOC arms, respectively (HR: 0.60; 95% CI 0.45–0.80; two-sided *p* = 0.0004) leading to the FDA approval of pembrolizumab in June 2020 for the first-line treatment of metastatic or unresectable MSI-H CRC [72,73,74]. Another cohort of the CheckMate 142 trial evaluating nivolumab plus low-dose ipilimumab in the first-line therapy reported an ORR of 69% (95% CI, 53 to 82) with 13% complete response rate. Median PFS and median OS were not reached with minimum follow-up of 24.2 months (24-month rates, 74% and 79%, respectively) [73]. These encouraging findings pave the way for additional phase III trials (NCT04008030, NCT02997228) currently investigating the added value of combined anti-CTLA or chemotherapy and targeted therapies to an anti-PD-1-L1 for the first-line treatment of MSI-H mCRC patients.

In the NICHE phase I/II trial [75], the effect of neoadjuvant immunotherapy by doublet ICIs (one single dose of ipilimumab and two doses of nivolumab 6 weeks prior to surgery) was investigated in a cohort of 40 patients with operable CC. Both MSI-H (21 patients) and MSS (20 patients) cancers were included, of which 35 were evaluable for efficacy (20 MSI-H and 15 MSS). The treatment was well tolerated, and all patients underwent radical resections without delay. Pathological response was observed in 20/20 of MSI-H tumors, with 19 major pathological responses (defined as ≤10% residual viable tumor on histopathology) and 12 (60%) pathological complete response (CR). The NICHE study data indicate that neoadjuvant immunotherapy has the potential to become the SOC for defined groups of CC patients when validated in larger studies. These study results are corroborated by early reports in rectal MSI-H cancer [76]. Several ongoing trials currently evaluate the benefit of ICIs combined or not with chemotherapy in the neoadjuvant and adjuvant settings of MSI-H CRC [77].

### 3.2. ICIs in MSS CRC

The clinical benefit of ICIs was observed for MSI-H CRC while the vast majority of MSS CRCs did not respond to this treatment. This could suggest that ICIs alone would not be sufficient to treat MSS tumors. Initial studies reported that only a very low proportion of MSS chemo refractory mCRCs benefit from anti-PD-1 combined or not with anti-CTLA-4. No MSS mCRC patients enrolled in the KEYNOTE-028 or KEYNOTE-016 trials responded to pembrolizumab treatment [78]. In CheckMate 142, only one MSS CRC patient achieved a partial response from a combination of nivolumab and ipilimumab [73]. The randomized phase 2 CCTG CO.26 trial suggested that combined immune checkpoint inhibition with durvalumab (anti-PD-L1) plus tremelimumab (anti-CTLA-4) may be associated with prolonged OS in patients with advanced refractory mCRC compared to best supportive care (HR: 0.72; 90% CI, 0.54–0.97; *p* = 0.07) [79]. Elevated plasma TMB (≥28 mut/mb, 21% of MSS mCRC) may select patients most likely to benefit from durvalumab and tremelimumab treatment. Interestingly, among the MSS tumors included in the NICHE trial [75], 4 of 15 had pathological responses (3 major and 1 partial response). The difference in response between MSS and MSI-H cancers was mainly attributed to a difference in TMB, MANA, and T cell infiltration. Notably, CD8+PD-1+ T cell infiltration was predictive of response in MSS tumors, suggesting that some MSS tumors are immune-responsive despite not demonstrating MSI-H at the molecular level.

As MSI-H tumors, *POLE/POLD1* CRC are characterized by high TMB and are potentially highly infiltrated by TILs. For this reason, treatment of *POLE/POLD1* mutant with ICIs is currently under investigation but is limited by the very low frequency (around 1%) of *POLE/POLD1* mutant in the CRC population. Recently, in a multi-national trial, five out of seven enrolled patients with *POLE/POLD1* mutant CRC achieved a clinical response to nivolumab in monotherapy [80].

Together, these findings suggest that a combination of different ICIs has marginal efficacy in MSS CRC but, most importantly, it underlies the lack of clear biomarkers, except for *POLE/POLD1* mutation, that could help to select MSS tumor subtypes that would be prone to respond to ICIs. Novel strategies are developed under the rationale of overcoming immune resistance and developing an effective immune response against tumor cells, such as combined strategies of immune checkpoint inhibition, immunotherapy-based combinations with chemotherapy and targeted therapy, radiation therapy, vaccines, and intratumoral strategies such as oncolytic viruses and bispecific antibodies. These numerous approaches are currently being evaluated in clinical trials [45,81].

## 4. Integration of Biomarkers of Immune Response and Resistance for the Development of Clinical Research Strategies for MSS CRC Immunotherapy

This section specifically discusses the current knowledge and clinical research strategies integrating predictive biomarkers of response and resistance to immune therapy (Figure 3) together with combined treatment able to overcome this CRC immune resistance. For easier comprehension, we describe, here, separately each biomarker with reported clinical efficacy and research development. However, all these markers are often linked and represent, as already highlighted, one of the pieces of the immune reactive pathway of CRC.

### 4.1. PD-1/PD-L1 Expression

PD-L1, expressed, among others, on tumor cells, can bind to its ligand, PD-1, expressed at the cell surface of activated T cells, NK cells, and B-cells [82]. Over the last few years, PD-L1 expression, evaluated by Immunohistochemistry (IHC) has been extensively evaluated as a predictive biomarker of response to ICIs in several solid cancers such as gastric cancer, esophageal tumors, and Non-Small-Cell Lung Carcinoma (NSCLC) [83]. However, the expression of PD-L1 is well recognized as a dynamic process and may vary according to TME, treatment, and stage of the disease. Moreover, PD-L1 expression is also induced by constitutive oncogene activation and Interferon-γ (IFN-γ), produced by activated lymphocytes [84,85]. Therefore, the assessment of PD-L1 expression by IHC is highly dependent on spatial heterogeneity and sampling. Several primary antibodies and staining conditions can be used for PD-L1 detection, thus inducing heterogeneity between laboratories. The image analysis and threshold used for quantitative detection on tumor and immune cells are also often different. Altogether, this biological and technical heterogeneity limits the use of PD-L1 as a predictive biomarker and efforts to harmonize PD-L1 staining and image analysis need to be made [86].

In CRC, PD-L1 expression was poorly correlated with MSI-H status [87] and was not found to be associated with response or survival in the registration studies [88]. A recent meta-analysis revealed that PD-L1 expression can serve as a significant biomarker for negative prognosis that is not related to clinicopathological characteristics [89]. Another meta-analysis reported that PD-L1 expressed on immune cells was associated with good prognosis, while PD-L1 expression on tumor cells has heterogeneous outcomes and does not meet requirements of a prognostic marker due to absence of standardization [90]. Few studies evaluating anti-PD-1/PD-L1 treatment in MSS mCRC reported PD-L1 expression on tumor samples. In the randomized phase III trial evaluating atezolizumab (anti-PD-L1) combined or not with cobimetinib (anti-Mitogen-Activated Protein Kinase 1 (MEK)), the ORR (3%) was not associated with PD-L1 expression [91].

Cytotoxic lymphocytes (CD8+) expressing PD-1, characterized by a memory/exhausted transcriptome, suggesting an antitumor T cell repertoire, are leader actors of the T-cell-mediated antitumor immunity [92]. In NSCLC, cytotoxic PD-1 high cell infiltration has been associated with clinical response to anti-PD-1 [93,94]. In MSS mCRC, it has been shown that cytotoxic PD-1 high infiltration without Th17 infiltration in tumors that express PD-L1 has a TME similar to MSI-H CRC and was associated with anti-PD1 benefit [92], as already reported in melanoma and digestive cancers [93].

Some ongoing studies (Table 2) with ICIs treatments for mCRC are currently investigating PD-1/PD-L1 expression, T cell proportions, and gene expression on blood samples or serial tumor biopsies as a dynamic biomarker (Table 2). One ongoing trial evaluates the combination of pembrolizumab together with favezelimab (anti-LAG-3) in PD-L1-positive mCRC (NCT05064059). LAG-3 (CD223) is another immune checkpoint molecule expressed at the cell surface of activated T lymphocytes, NK cells, B-lymphocytes, and plasmocytoïd dendritic cells which binds on the class II major histocompatibility complex (MHC) receptor.

### 4.2. POLE/POLD1 Mutation

In CRC patients, the application of *POLE/POLD1* mutation as a molecular marker for ICI treatment is being researched. In most cancers, the TMB of patients who carried *POLE/POLD1* mutations was significantly higher than that of non-carriers. Among patients treated with ICIs, the OS of patients who carried *POLE/POLD1* mutations was significantly better than that of non-carriers [95]. This study also found that 26% of the patients who had *POLE/POLD1* mutations also showed the MSI-H phenotype. After removing this subset of patients, the remaining patients with MSS were also found to benefit significantly from ICIs treatment. Multivariate Cox regression analysis showed that *POLE/POLD1* mutation was an independent factor determining which solid tumor patients may benefit from ICI treatment. At present, there are several clinical studies focusing on *POLE/POLD1* mutation and ICI treatment (Table 3), and more evidence supporting the use of POLE/POLD1 mutation as molecular markers is expected.

### 4.3. Tumor Mutational Burden

As MSI-H tumors harbor a high number of mutations and a high MANA load, the TMB, expressed as the number of non-synonymous somatic mutations per trillion bases, emerged as a predictive biomarker of response to ICIs. In CRC, TMB-High (TMB-H) correlates with MSI-H and is associated with a high MANA load and immunogenicity [96]. It has been reported that 97% of MSI-H are TMB-H, as defined by a cutoff of 10 mutations per megabase [97]. However, only 16% of TMB-H are MSI-H, suggesting that MSS TMB-H CRC is more common than MSI-H CRC and could benefit from ICIs [98,99]. In KEYNOTE-158, enrolling non-CRC MSI-H patients, a correlation between the antitumor activity of the anti-PD-1 pembrolizumab and TMB-H has been reported. The ORR of TMB-H tumors was 29%, while the ORR of non-TMB-H tumors was 6% [100]. Following these observations, the FDA approved pembrolizumab for the treatment of TMB-H refractory advanced solid tumors, highlighting the promising value of TMB as an independent predictive biomarker of response to ICIs. Moreover, in a recent report, Fabrizio et al. found that the ability of TMB to identify the CRC subgroup of patients that may respond to ICIs outperformed that of MSI status [101].

However, as a pan-cancer marker, a fixed cutoff of TMB that can be applied to different tumors was difficult to identify [99]. In a meta-analysis, the researchers summarized the most common cutoff values of TMB, which were 10, 16, and 20 mut/Mb. Schrock et al. reported that for 22 MSI-H mCRC patients treated with PD-1 or PD-L1 inhibitors, the optimal cutoff value of TMB associated with better outcomes was between 37 and 41 mut/Mb [102]. Another trial evaluating regorafenib and nivolumab treatment in chemo refractory MSS mCRC (REGONIVO trial) reported an optimal TMB cutoff value of 22.55 mut/Mb for OS benefit [103]. Elevated plasma TMB (≥28 mut/Mb) may select patients most likely to benefit from durvalumab and tremelimumab treatment in the phase 2 CCTG CO.26 trial [79]. This suggests that a unique and optimal TMB high threshold does not exist for all cancers but more so, it could differ within different molecular subgroups of tumors as highlighted here for CRC.

The methodology for TMB evaluation is also an important characteristic to consider. Whole-Exome Sequencing (WES) is the gold standard for TMB assessment, but this technique is expensive and lacks uniformity [104]. Next-Generation Sequencing (NGS) is a widely used and cheaper method regarding its convenience and applicability, but it introduces bias and errors related to the used panel size [105]. A correlation between TMB predicted by NGS and WES is reported [106]. Nevertheless, the methods used to calculate TMB in NGS and WES also affect the TMB results. Blood-based TMB is currently a valuable substitute for tissue TMB because of its facility in sampling and high consistency with tissue TMB in the predicted results [107]. However, assessment of TMB on circulating tumor DNA (ctDNA) requires the sequencing of a large panel of genes and, ideally, high coverage is also required to be able to distinguish an increase of TMB from an increase of ctDNA. If these conditions are not fulfilled, the assessment of TMB on blood sample could be very volatile and difficult to interpret correctly.

Several trials are currently recruiting CRC patients harboring high TMB tumors to be treated by ICIs-based therapies. One trial (NCT04695470) is combining sintilimab (anti-PD-1) with fruquitinib (VEGFR-1, -2, -3 inhibitor) for the treatment of refractory MSS mCRC with high TMB. In this trial, the TMB is assessed by NGS on plasma samples (TMB ≥ 5 mutations/Mb). Another trial (NCT03638297) recruiting CRC patients with MSI-H or high TMB tumors evaluates the association of a Cyclooxygenase (COX) inhibitor (BAT1306) with pembrolizumab treatment. Studies with preclinical models reported that COX inhibitors could act with PD-1 antibody in mice and control disease progress. In addition, COX-2 could drive constitutive expression of IDO1 in human tumor cells. This could contribute to overcoming the lack of T cell infiltration and render the tumor more immunogenic [108].

### 4.4. DNA Methylation

O6-Methylguanine–DNA Methyltransferase (MGMT) is a key protein in the DNA repair mechanism of damages induced by alkylating agents and, as illustrated in glioblastoma, the epigenetic silencing of MGMT is a mechanism that potentiates the effect of Temozolomide (TMZ) [109]. Despite the fact that MGMT methylation, inducing a lack of MGMT protein expression, is found in 40% of CRC patients, TMZ and its analog dacarbazine provided limited clinical activity with an ORR under ten percent in MGMT-methylated mCRC [110]. As demonstrated in CRC models and mCRC patients [111], resistance to TMZ, observed in almost all TMZ-sensitive tumors [112], may be related to the hypermutated phenotype and the emergence of other mutations in DNA repair mechanisms such as MSH6 [113]. In the proof-of-concept trial MAYA, designed to assess the potential role of TMZ as an inducer of hypermutated phenotype and immune-sensitizing agent in MSS MGMT-silenced mCRC tumors, eligible patients received two cycles of TMZ followed by its combination with anti-CTLA-4 ipilimumab in the absence of progression [114]. In this trial, among the 204 eligible patients, 135 started TMZ treatment and 102 of them were discontinued due to disease progression or death. Among the 33 patients who achieved disease control and received ipilimumab, 36% reached the primary objective of eight-month PFS rate. The ORR was 45% and the median OS and PFS were 18.4 and 7 months, respectively. This proof-of-concept trial provided new insight regarding the strategy of turning cold tumors into hot tumors through the use of TMZ inducing hypermutated phenotype. However, further investigation on treatment regimen, optimization, and patient selection is needed in order to maximize the success of this therapeutic approach. Similar to the MAYA trial, the ARETHUSA (NCT03519412) trial is currently investigating the efficacy of pembrolizumab in patients that reach >20 mutations/Mb after TMZ priming. Moreover, another group is currently conducting a phase Ib trial (NCT04689347) combining fluorouracil, leucovorin, and irinotecan/bevacizumab (FLIRT/BEV) with escalating doses of TMZ in untreated MSS MGMT-silenced mCRC in order to investigate the optimal dosing of this new triplet and the role of maintenance immunotherapy in patients with disease control after the FLIRT/BEV regimen.

Previous studies have demonstrated that epigenetic modulation by DNMT inhibitor modifying the expression of genes related to innate immunity, adaptive immunity, and immune evasion in tumor tissues [115,116,117] may enhance the antitumor immune response by promoting increased TILs, although specific mechanisms by which this occurs have not been established [118]. In this way, a phase 2 single-arm trial has evaluated activity and tolerability of pembrolizumab plus azacytidine (DNMT inhibitor) in patients with chemotherapy-refractory mCRC [119]. This treatment combination provided modest activity (ORR: 3%, median PFS: 1.9 months, median OS: 6.3 months); correlative studies suggest that tumor DNA demethylation and immunomodulation occur. While not sufficient for antitumor activity, this immunomodulatory approach may contribute to future strategies to overcome immune resistance in patients with mCRC.

### 4.5. Gene Expression Profile and Consensus Molecular Subtypes

As discussed, the CMS1 and CMS4 groups are both characterized as CRC infiltrated by immune cells and could be potential predictive biomarkers. However, their immune environment is very different and could be summarized as immunoreactive for CMS1 and immunosuppressive for CMS4. If CMS1 tumors have a TME which may benefit from ICIs, the situation could be more complex for CMS4 CRC. The TME of these tumors with the presence of Tregs, MDSC, M2 macrophages, Th17 cells, and IFN-γ signature suggests the potential for immune therapy benefit but with additional efforts to overcome this immunosuppressive microenvironment [16,37,46].

To our knowledge, no ongoing clinical trials directly use CMS1 subtype as a predictive biomarker of response to ICIs. This is mainly explained by the fact that the CMS1 subgroup often comprises other recognized molecular biomarkers such as MSI-H status, *POLE/POLD1* mutation, and high TMB which are easier to assess. Interestingly, a recent phase 2 study [120] evaluated the combination of encorafenib (BRAF inhibitor), cetuximab (anti-Endothelial Growth Factor Receptor (EGFR) monoclonal antibody), and nivolumab in patients with MSS, *BRAF-V600E*-mutated mCRC, a mutation frequently associated with CMS1 subgroup. Preclinical models of MSS CRC showed that BRAF combined with EGFR inhibition induced a transient MSI-H phenotype [120]. *BRAF V600E* inhibitor may prime these tumors for response to anti-PD-1 antibodies. In this trial, the 26 enrolled patients experienced an ORR of 45% with a median PFS of 7.3 months and OS of 11.4 months. A follow-up randomized phase II trial (SWOG 2107) to evaluate encorafenib/cetuximab with or without nivolumab in patients with MSS, *BRAF-V600E*-mutated metastatic CRC will begin in 2022.

CMS4 CRC features increased TGF-β signaling, which may account for de novo resistance to immunotherapy for patients with MSS mCRC. One recent phase 2 trial (NCT03436563) [121] evaluated bintrafusp alfa, a dual PD-L1 antibody/TGF-β trap, with radiation therapy to a single metastatic lesion with abscopal intent (8 Gy) for the treatment of CMS4 mCRC. No patients achieved a tumor response and median PFS and OS were 1.6 months and 5.0 months, respectively. Although the efficacy for bintrafusp alfa and radiotherapy was deceiving, changes in IFN-γ signature provide a potential signal for refining therapeutic strategies based upon TGF-β enrichment in patients with mCRC. Another cohort from the same trial (NCT03436563) [121], focusing on MSI-H cancers refractory to ICIs did not demonstrate significant antitumor activity. Ongoing correlative studies may inform on the effect of TGF-β and PD-L1 modulation by bintrafusp alfa within the TME. Another phase I/II trial investigated the efficacy of avelumab plus the autologous dendritic cell vaccine in pretreated MSS mCRC patients [122]. An interim analysis (Simon design first-stage) recommended early termination because 11% (2/19) of patients were progression-free at 6 months and no patients experienced tumor response. Four patients (21%) experienced hyper-progressive disease. Stimulation of immune response was observed with changes of cytokine levels after treatment. The evaluation of transcriptomic immune-metabolic signature did not correlate with clinical outcomes. Hyper-progressive disease was observed in different immune-metabolic micro-environments.

Although CMS classification appears deceiving for biomarker development, ongoing trials focusing on different compounds (TGF-β, Tregs, MDSCs…) of the immune suppressive TME enriched in CMS4 CRC are still ongoing. 

### 4.6. Tumor-Infiltrating Lymphocytes and Immunoscore

Tumor infiltration of cytotoxic T cells and Th1 cells and IFN-γ upregulation predict a favorable prognosis in CRC [123] and also serve as markers indicating a good response to Immune Checkpoint Inhibitors [124], as IFN-γ can upregulate PD-L1 and Major Histocompatibility Complex-I (MHC-I) expression by tumor cells [125]. In addition, co-localization of PD-L1+ cells with tumor-infiltrating CD8+ T cells has been widely reported as a predictive biomarker for ICI treatment [126,127,128]. The hypothesis is that TILs induce adaptive immune resistance, accompanied by increased PD-L1 expression. Indeed, CD8 and PD-L1 expression is significantly higher in responders to anti-PD-1 therapy than in non-responders [126,129,130].

Immunoscore has been developed, extensively evaluated in several cohorts of patients, and recognized as a robust prognostic biomarker [131]. However, its use as a predictive biomarker is still under investigation. Moreover, the use of Immunoscore on biopsies is difficult since it limits the analysis to a restricted part often limited to the core of the tumor. In this way, the prognostic and the predictive values of a biopsy-adapted Immunoscore (IS_B_) were evaluated in a recent study. Pre-therapeutic biopsies from patients with locally advanced rectal cancer treated with CRT followed by radical surgery were stained for CD3+ and CD8+ T lymphocytes in two independent cohorts. Density of CD3+ and CD8+ T cells was used to derive an IS_B_ [132]. Authors reported that a high IS_B_ positively correlated with cytotoxic immune response, Th1-orientated gene expression signature, and histologic response after treatment. In addition, patients with high IS_B_ were at lower risk of relapse or death compared with low IS_B_. Today, some trials continue to investigate Immunoscore as a prognostic biomarker (NCT04938986, NCT01688232, NCT0342260, NCT02274753) for disease-free survival stratification and detection of risk of recurrence (Table 4).

Recent studies suggest using Immunoscore to predict the response to immunotherapy [38,45,133], but besides Immunoscore, TILs and immune cell populations in the tumor could also be used as prognostic or predictive biomarker. The POCHI trial (NCT04262687) is currently recruiting MSS mCRC with high immune infiltration evaluated by Immunoscore, among others, in an experimental arm combining XELOX plus bevacizumab and pembrolizumab (Table 4). This proof-of-concept study could pave the way for the use of Immunoscore as a predictive biomarker of ICIs soon.

Beyond CD3+ and CD8+ T cells, other immune cells within the TME, such as Th17 and memory T cells, have gained interest in the past years. Th17 cells, secreting IL-17, modulate the expression of other cytokines such as TGF-β, IFN-γ, IL-6, IL-21, and IL-22. The role of Th17 cells in tumor immunity and development remains controversial, mainly attributed to the plasticity of Th17 cells. In CRC, Th17 seems to play a role in carcinogenesis and could decrease the antitumor activity of CD8+ T cells [134]. Intratumoral IL17-mediated signaling may preclude responses to immunotherapy. A recent paper reported that both IL-17 low and high immunoreactive MSS CRC are associated with features of adaptive immunosuppression (CD8/IFN-γ and PD-L1/IDO1 co-localization). Nevertheless, only patients with a Th17 low MSS CRC had a TME resembling that of patients with mCRC responsive to anti-PD-1 treatment [92]. Several studies reported that the presence of memory T cells (CD45RO+) within the TME was associated with better prognosis and lower risk of CRC metastasis [50,135,136,137]. CD8+ resident memory T cells are found in greater abundance in MSI-H CRC, suggesting an important role in the antitumor immunogenicity of MSI-H CRC [138]. Their presence in MSS CRC could be an additional marker suggesting the immunoreactiveness of the tumor and the possible response to immune therapy.

### 4.7. The Gut and Cancer Microbiome

It is now well established that gut microbiome is strongly involved in the development and maintenance of the host immune system. In this regard, seminal papers have highlighted different responses to ICIs in cancer patients depending on the composition of their gut microbiome [139,140]. For instance, specific bacterial species have been associated with better prognosis and response to anti-PD-1 ICI in melanoma patients [141,142,143,144]. In NSCLC, the authors reported that *Akkermansia muciniphila* (*Akk*) was associated with increased ORR and survival after anti-PD1 ICI [145]. Another study recruiting patients with gastrointestinal cancer found a significant different *Prevotella/Bacteroides* species ratio associated with ICIs responses [146]. A potential mechanism by which the distant microbiota might benefit from tumor immunotherapy is through bacterial metabolites. Recently, authors report that a collection of eleven bacteria from human gut microbiota appeared to be able to robustly induce IFN-γ, producing CD8+ T cells in the gut and enhancing antitumor immunity [147]. Furthermore, Mager et al. reported, in mouse models, that the metabolite inosine derived from intestinal populations of *Bifidobacteria, Lactobacillus*, and *Olsenella* was associated with increased numbers of CD8 +IFN-γ + T cells and control of tumor growth [148].

Several ongoing trials currently study the gut microbiome. The NCT02960282 trial studies the gut microbiome in fecal samples from mCRC patients treated with chemotherapy or immunotherapy. An upcoming phase II study (NCT05279677) will evaluate the efficacy and safety of fecal microbiota transplantation plus sintilimab and fruquitinib in chemo refractory mCRC.

Beyond the gut microbiome, a diverse microbial community is also present within the CRC. Among them, *Fusobacterium nucleatum* (*Fn*), a tumor-resident bacteria in CRC, is of growing interest. Several publications have reported correlations between intra-tumoral detection of *Fn* and poor prognosis, shorter PFS, and higher tumor recurrence [60,149,150,151]. In addition, a recent publication reported that high *Fn* levels correlated with better therapeutic response to PD-1 blockade in CRC patients, regardless of MSI status [152]. Furthermore, *Fn* induced PD-L1 expression by activating Stimulator Of Interferon (STING) signaling and increased the accumulation of IFN-γ+ CD8+ TILs during treatment with PD-L1 blockade, thereby increasing tumor sensitivity to PD-L1 blockade. Due to its recent discovery and the lack of sufficient knowledge, there are currently no ongoing trials evaluating the cancer tissue microbiome as a biomarker of response to ICIs.

### 4.8. Circulating Biomarkers

mCRC is characterized by the important inter- and intra-patient heterogeneity of its intratumoral immune microenvironment [57]. Even though Immunoscore, gene expression profiles, or TMB have predictive or prognostic value [153], their practical information is limited by tumor tissue accessibility and spatial and temporal heterogeneity during the CRC evolution. Due to the tumor’s dynamic behavior, the study of tumor characteristics requires the evaluation of accessible biomarkers that can reflect tumor modifications during treatment without the need for biopsy. Therefore, identifying biomarkers in body fluids that can accurately, quickly, and cost-effectively reflect the stage and characteristics of the tumor is desired. Circulating exosomes, microRNAs, tumor cells (CTC), and tumor DNA can be ideal indicators for tumor heterogeneity changes during treatment [154]. The similarity of the genetic profile of CTCs with tumors has been reported in 50–77% of cases [155,156]. Interestingly, the genetic profile similarity between cell-free DNA (cfDNA) or ctDNA and tumors has been reported to be more than 90% [157,158].

New liquid biomarkers have been recently investigated. Circulating tumor DNA, flow/mass cytometry, and blood T Cell Receptor sequencing (TCR-seq) allow sensitive tracking of changes in antigen-specific T cells at the clonal level, with unprecedented insight into mechanisms by which ICIs alter T cell responses [159]. Serial sampling and combination of these approaches will likely become a key element to provide an overview of the genetic makeup of the tumor and adaptive immunity of the patient.

#### 4.8.1. Circulating Tumor DNA

Commonly designated as “liquid biopsies”, the analysis of the ctDNA by NGS or digital droplet PCR (ddPCR) is currently extensively investigated in several cancers [160]. Liquid biopsies have emerged as a diagnostic tool to assess the presence of tumoral mutations and to monitor the emergence of mutations over time. In CRC, the assessment of ctDNA alteration of genes belonging to the EGFR pathway predicted response to anti-EGFR treatment [158]. The emergence over time of such alteration during anti-EGFR treatment is associated with acquired resistance [161]. The MSI-H status as well as the TMB can also be assessed on ctDNA with similar predictability for response to ICIs [162]. Detection of ctDNA in the follow-up of the treated patient is also used to detect minimal residual disease and often reveals earlier recurrence compared to standard radiology [163].

A study including 1000 patients with advanced or metastatic tumors treated with ICIs reported that on-treatment ctDNA dynamics appear to be predictive of the long-term benefit of ICI across tumor types. ctDNA dynamics could help to select patients who will ultimately derive benefit from immunotherapy [164]. A prospective clinical trial [165] also revealed the correlation between the level of ctDNA and the efficacy of ICI treatment in five different cancer types. Authors measured the ctDNA level after three cycles of pembrolizumab and found that patients with a decrease of ctDNA level showed better clinical efficacy during the treatment. A study of MSS CRC patients treated with regorafenib and PD-1 inhibitors found that ctDNA may be predictive of early therapeutic efficacy. Specifically, ten patients with rising ctDNA levels or emergence of new clones four weeks after treatment experienced progressive disease after two months, whereas three patients with declining ctDNA experienced stable disease [166].

Several ongoing clinical trials currently use ctDNA to select molecular subgroups of CRC (MSI-H, TMB-H, or specific mutation of interest) more easily than tissue biopsy (Table 5). ctDNA is also used to map precise disease evolution; dynamic change in ctDNA is measured to detect response and early resistance to ICIs (NCT03946917, NCT04046445, NCT05240950, and NCT02997228).

#### 4.8.2. T Cell Receptor Repertoire

The specificity of the tumor immune response is linked to the antigen-specific T Cell Receptor (TCR). The analysis of the TCR repertoire on Peripheral Blood Mononuclear Cells (PBMCs) and TILs could be used as a prognostic and predictive biomarker of response to ICIs [167]. T cell receptors recognize specific antigens presented by MHC class I (for CD8+ T cells) or MHC class II (for CD4+ T cells) molecules. A process called genetic recombination occurs in T cells to rearrange the DNA at three T Cell Receptor Beta Loci (TRBV, TRBD, and TRBJ) to develop TCRs that are specific for certain antigens. The T cell repertoire refers to all of the unique TCR genetic rearrangements within the adaptive immune system; thus, the TME T cell repertoire refers to all of the unique TCR genetic rearrangements within the TME. Not surprisingly, having a diverse TME T cell repertoire is associated with better outcomes in response to immunotherapy [159]. As for liquid biopsies, TCR repertoire can be assessed pre-treatment but also over time to measure dynamic changes of the repertoire in response to ICIs. The TCR repertoire analysis can be performed on all PBMCs found in the blood (bulk TCR) or performed on specific subsets of lymphocytes sorted by flow/mass cytometry such as CD8+PD-1+ T cells that are the main target of PD-1 blockade [159]. In melanoma, it has been shown that a high pre-treatment TCR diversity of blood CD8+PD-1+ T cells and a reduced post-treatment diversity are associated with a longer PFS after anti-PD-1 therapy [168,169]. Still in melanoma, single cell analysis of peripheral CD8+ T cells revealed that responders to both anti-PD-1 and anti-PD-1 + anti-CTLA-4 presented more expanded clones than non-responders. Similarly, it has been shown that after one cycle of ICIs, responders exhibited clonally expanded CD27+ C-C Motif Chemokine Receptor 7 (CCR7)+ memory T cells. How TCR diversity impacts adaptive immunity in CRC patients remains unclear for ICI treatment. Very limited data on TCR repertoire and ICIs are available for CRC. One report [170] showed that mCRC patients treated with chemotherapy and with either high baseline TCR diversity or TCR diversity that dropped during therapy had significantly better treatment responses. In a TCR repertoire analysis of advanced CRC patients treated with a combination of five cancer peptide vaccines and oxaliplatin-based chemotherapy, high TCR diversity scores were associated with improved response [171]. The expansion of tumor-associated TCRs in the blood underscores the continuity of the tumor and blood compartments and suggests that the activity of PD-L1 blockade may involve circulating T cells more than previously thought. It raises the possibility that antitumor T cells may home from the periphery into the tumor before later recirculating. In addition, a recent report suggested a significant difference in the usage of TRBV and TRBJ genes between CRC patients and healthy controls, supporting its use as an additional TCR-based predictive biomarker in CRC [172].

TCR repertoire is currently investigated in several CRC trials involving ICIs (Table 5). In one trial (NCT03927898) combining toripalimab (anti-PD-1) together with stereotactic body radiotherapy for the treatment of oligometastatic CRC, investigators will analyze dynamic TCR repertoire changes in peripheral blood as well as PD-1 and Ki67 expression on T cells and PD-L1 expression on circulating tumor cells. Moreover, in another window of opportunity trial (NCT04714983), the immunotherapeutic response after OX40-ligand expressing oncolytic adenovirus will be evaluated by TCR repertoire changes in blood and tissue samples.

#### 4.8.3. Flow/Mass Cytometry

Flow cytometry and mass cytometry (allowed to cover up to 40 markers) can provide interesting data on the frequency of T cells subpopulations and their variation during treatment. Flow or mass cytometry permit one to obtain a comprehensive overview of tumor-resident and circulating immune populations. Several simultaneous staining panels (e.g., CD45, CD3, CD8, CD4, CD11B, CD14, CD20, CD25, FOXP3, PD-1, TIGIT, …) can be performed, allowing a global characterization of the circulating immune subpopulations and in the TME [173]. In melanoma and Merkel cell carcinoma, it has been shown that an increase of the frequency of PD-1+TIGIT+CD8+ circulating cells after one month of anti-PD-1 therapy was associated with OS and clinical response [174]. In another trial on melanoma, authors demonstrated that patients who respond to anti-PD-1 treatment showed a decrease of circulating PD-1+ Tregs (CD4+CD25+CD127-) [175]. In CRC, Th1 (CD126-CD4+) cells were more abundant in the blood of patients responding to anti-PD-1 compared with non-responders [176]. Pre-treatment Tregs frequency was higher in non-responders. Moreover, a decrease of the frequency of Th1 cells during the treatment was observed in patients with acquired resistance to treatment. The analysis of blood Th1 cells together with Tregs could represent a blood biomarker of response to ICIs in CRC.

The relative distribution of immune cells subtypes and immune checkpoint molecules expression on tumor cells and immune cells, assessed by flow cytometry in blood and/or tissue samples, is currently investigated in several trials (NCT05086692, NCT03984578, NCT02851004) combining pembrolizumab with other treatments (chemotherapy, cetuximab, STAT3 inhibitor). Another trial (NCT05086692) also using flow cytometry, evaluates a treatment with MDNA11, a drug designed to bind IL-2 beta receptor on immune cells, combined or not with ICI (Table 5).

Flow cytometry is also heavily used in clinical trials involving Chimeric Antigen Receptor (CAR) T cells. This tool allows the investigator to follow in vivo, in peripheral blood, the rate of CAR T cells in a dynamic way to evaluate the proliferation and long-term survival of the cells during the therapy (Table 5). CAR T cells are cells that have been genetically engineered to produce artificial chimeric T cell receptors [177]. CAR T cells have demonstrated great success in treating CD19-positive B cell malignancies [178]. Today, CAR T cells are also investigated in solid tumors such as CRC, either targeting, among others, Carcinoembryonic Antigen (CEA) (NCT04348643, NCT02349724, NCT04513431) or tyrosine-kinase Met (c-Met) (NCT03638206). 

#### 4.8.4. Cytokines

Cytokines play a key role in both pro- and antitumor immune responses and can be secreted by either the tumor cells or immune cells [179]. The cytokine A Proliferation-Inducing Ligand (APRIL), produced by tumor cells, and B-cell Activating Factor (BAFF), IL-8, and Matrix Metallopeptidase 2 (MMP2), produced by a variety of tissue and blood cells, have been reported to be inversely correlated with immune cell infiltration and expression of CD163, a marker of M2 macrophages in CRC [180]. Authors reported that the significantly decreased APRIL and increased BAFF, IL-8, and MMP2 expression was tumor-specific and deserves consideration in the development of new treatments [180]. Another study revealed, in MSS CRC cell lines and tissues, that IL-17A, secreted by Th17 cells, increased expression of PD-L1 on CRC cells and that inhibition of IL-17A improved the efficacy of anti-PD-1 therapy in a murine MSS CRC model [181]. IL-17A might serve as a therapeutic target to sensitize patients with MSS CRC to ICI therapy. Nevertheless, further investigation is needed to use cytokine as a biomarker of selection in clinical research.

Cancer-Associated Fibroblasts (CAFs) are cells within the TME promoting tumorigenic features by initiating the remodeling of the extracellular matrix or by secreting cytokines such as TGF-β, IL-6, IL-8, CXCL14, CXCL12, and VEGF [182]. Therefore, CAFs may act on cancer development, metastasis process, and tumor immunity. However, due to the transcriptional and functional heterogeneity of the CAFs, their use as therapeutic targets or biomarkers is still controversial and further investigation is needed to better understand their complexity [183].

### 4.9. Clinical Tumor Burden, Metastases Location, and Characteristics

Accumulating evidence suggests that a high tumor burden, defined as the total amount of tumoral tissue in the body estimated by imaging, liquid biopsy (circulating tumor DNA or cells), or biological tumor derivatives (e.g., Lactate Dehydrogenase (LDH), CEA) is negatively correlated with antitumor immunity and ICI response [184,185,186,187]. An enlarging tumor implies inefficacy of the immune system at containing its growth, while smaller tumor burdens may be partly immune-controlled.

Beside tumor burden, the impact of Liver Metastases (LM) on ICI effectiveness appears important. Tumeh et al. initially reported that PD-1 blockade was much less effective in melanoma or NSCLC patients with LM (ORR and median PFS of 47.5% and 5.1 months for patients with LM compared to 70.8% and 20.1 months for those without LM) [188]. This observation was subsequently confirmed by other clinical reports [189,190,191]. A proposed mechanism of this “systemic immune suppression” involves CD11b+ suppressive macrophages generated in and that delete CD8+ T cells through Fas Ligand [191]. In addition, Treg activation (CTLA-4, PD-1, and ICOS- high) was observed and contributed to the distal immunosuppression [192]. Preclinical modeling suggests that LM resection remodels systemic antitumor immunity by decreasing immunosuppression [193,194]. Correlative studies in patients undergoing resection of liver tumors have mainly focused on acute postoperative immunosuppression that underlies perioperative increased infection risk and have not evaluated the impact on the adaptive immune system [195]. However, the clinical benefits of resection are apparent. The resection of oligometastatic CRCLM in select patients has been associated with prolonged OS [196]. An ongoing clinical trial is examining the resection of CRCLM in combination with immunotherapy in CRC (NCT03844750). Radiofrequency ablation, cryotherapy, transarterial chemo- or radio-embolization, and stereotactic body radiotherapy are minimally invasive approaches that have demonstrated clinical utility in the management of CRCLM. Preclinical evaluations have shown that these treatments can induce an in situ vaccination which potentiates spontaneous and therapeutic antitumor immunity [193]. Many clinical trials are examining whether these ablative techniques of CRCLM are safe and efficacious in combination with immunotherapy.

Finally, several confounding clinical factors, such as sex/gender, age, obesity, race, alcohol and tobacco consumption, exercise, and psycho-emotional stress have been reported to possibly influence the magnitude of benefit to ICIs. All these factors may directly or indirectly act on the immune system, thus influencing ICIs response [197]. Therefore, all of these parameters should also be considered when analyzing the response to ICIs in CRC patients.

## 5. Conclusions and Future Direction

The current clinical use of the anti-PD-1 inhibitors pembrolizumab and nivolumab and the anti-CTLA-4 therapy ipilimumab has revolutionized the survival potential of many patients with MSI-H CRC and indicates the significant anticancer potential of ICIs for CRC. There are many areas of current research that aim to improve clinical response rates and generalize these treatments to all CRC. The immune microenvironment of MSS CRC is highly heterogeneous, which compromises the current development of immune therapies. Several genomic, transcriptomic, and TME classifications have been performed to distinguish and characterize the different immune subtypes of CRC. While immune-competent CRC often responds to ICI treatment, the situation remains more complex for immunocompromised or suppressed CRC, requiring innovative strategies based on immunotherapy, often combining ICI with chemotherapy, radiotherapy, or targeted therapies. These strategies aim at triggering an effective and adequate immune response against tumor cells.

The elucidation of the different pathways mediating immune resistance in MSS CRC remains an ongoing challenge for clinical research. Improving patient selection through effective biomarkers identifying the different immune subtypes of CRC remains essential for future research development. Although Immunoscore, gene expression profiles, or TMB have predictive or prognostic value, their practical application remains limited by the accessibility of tumor tissues and the spatial and temporal heterogeneity during CRC progression. Therefore, circulating biomarkers that can accurately, rapidly, and cost-effectively reflect the dynamic behavior of the tumor during treatment without the need for biopsy are of great interest for future research. Serial sampling and the combination of tissue and liquid biopsy approaches will likely become a key component in providing a comprehensive view of the genetic makeup of the tumor and the patient’s adaptive immunity.

The development of prospective clinical trials combining ICIs with other therapies and aiming at proving the predictive value of selected biomarkers is crucial. Such trials would help the clinical and scientific communities to better understand the complex interactions of the cancer cells with their surrounding TME and to optimally use new combination approaches. Finally, although MSI-H CRCs are known to respond to ICIs, not all benefit from these treatments. Therefore, the development of new combination approaches when ICIs resistance occurs and the use of biomarkers for understanding must continue. Hopefully, this will allow scientists and oncologists to exploit the full potential of immunotherapy in the treatment of CRC.

## Figures and Tables

**Figure 1 cancers-14-02241-f001:**
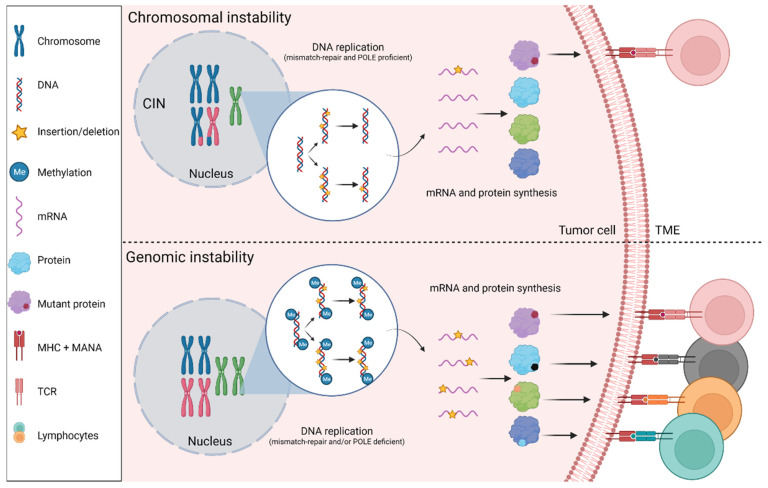
**Chromosomal Instability (CIN) versus genomic instability.** In CRC tumor cells, characterized by chromosomal instability, the majority of insertions and deletions (indels) occurring during DNA replication are repaired by a functional mismatch-repair mechanism and the DNA Polymerase Exonuclease Domain (POLE/POLD1). A few Mutation-Associated Neoantigens (MANAs) are presented to the cell surface by the Major Histocompatibility Complex (MHC) and recognized by lymphocytes. In CRC tumor cells characterized by genomic instability and CpG Island Methylator Phenotype (CIMP), the majority of indels occurring during DNA replication are not repaired. A high number of mutant proteins are translated, inducing a high number of MANAs presented and recognized by lymphocytes. mRNA: messenger RNA; TCR: T Cell Receptor; TME: Tumor Microenvironment.

**Figure 2 cancers-14-02241-f002:**
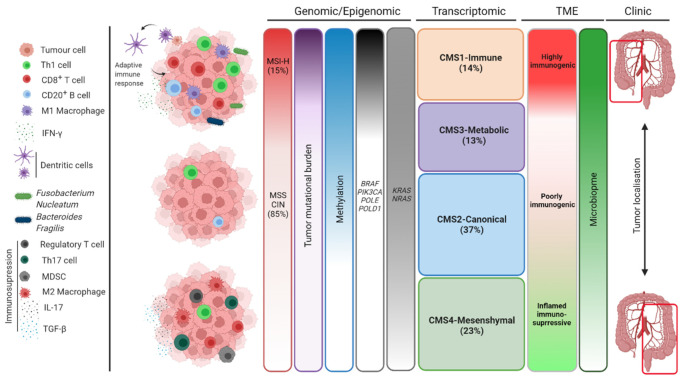
**Colorectal cancer classifications:** Th: T-helper lymphocytes; IFN: Interferon; MDSC: Myeloid-Derived Suppressor Cell; IL: Interleukin; TGF: Transforming Growth Factor; MSI: Microsatellite Instability; MSS: Microsatellite Stable; CIN: Chromosomal Instability; CMS: Consensus Molecular Subtype; TME: Tumor Microenvironment.

**Figure 3 cancers-14-02241-f003:**
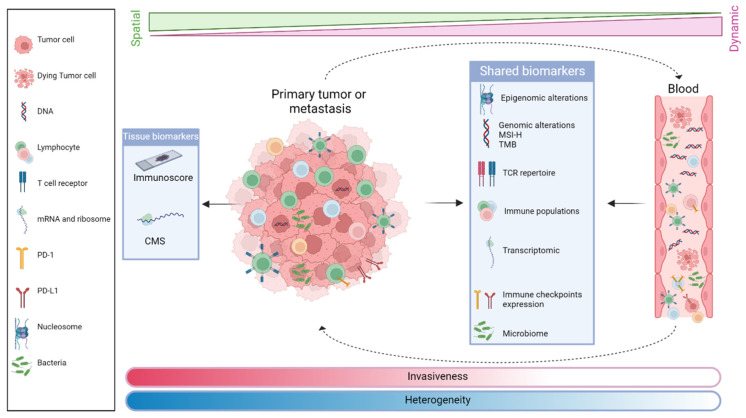
**Prognostic and predictive tissue and blood biomarkers of response to immune checkpoint inhibitor therapy:** PD-1: Programmed Cell Death protein 1; PD-L1: Programmed Death-Ligand 1; CMS: Consensus Molecular Subtype; MSI: Microsatellite Instability; TMB: Tumor Mutational Burden; TCR: T Cell Receptor.

**Table 1 cancers-14-02241-t001:** Selected ICIs trial results in MSI-H CRC.

Clinical Trial	Phase	Treatment	Setting	Primary Endpoints	OS	PFS	ORR	HR
KEYNOTE-016	II	Pembrolizumab	Refractory mCRC Cohort A: MSI-H CRC Cohort B: MSS CRC Cohort C: MSI-H non-CRC	ORR PFS	Median OS not reached (A, C); median OS of 5 months in cohort B	A: 78% B: 11% C: 67%	A: 40% B: 0% C: 71%	A vs. B (for death) (0.22; 95% CI 0.05–1.00; *p* < 0.001) A vs. B (for progression or death) (0.04; 95% CI 0.01–0.21; *p* < 0.001)
KEYNOTE-016 Update	II	Pembrolizumab	Refractory MSI-H cancers Cohort A: CRC Cohort B: non-CRC	ORR PFS	Median not reached yet	Median not reached yet	A: 52% B: 54%	NA
KEYNOTE-164	II	Pembrolizumab	MSI-H refractory mCRC Cohort A: ≥2 prior lines Cohort B: ≥1 prior lines	ORR	A: 55% (24 months) B: 63% (24 months)	A: 31% (24 months) B: 37% (24 months)	A: 33% B: 33%	NA
KEYNOTE-177	III	Pembrolizumab	Treatment naive MSI-H mCRC Cohort A: Pembrolizumab Cohort B: SOC	PFS OS	A: 61% (36 months) B: 50% (36 months)	A: 42% (36 months) B: 11% (36 months)	A: 69% B: 51%	OS: (0.74; 95% CI 0.53–1.03; *p* = 0.036) PFS: (0.61; 95% CI 0.44–0.83; *p* = 0.0008)
CheckMate 142	II	Nivolumab	Refractory MSI-H mCRC	ORR	73% (12 months)	50% (12 months)	31%	NA
CheckMate 142	II	Nivolumab + Ipilimumab Nivolumab	Refractory MSI-H mCRC Nivolumab (3 mg/kg) Ipilimumab (1 mg/kg × 4) Nivolumab (3 mg/kg every 2 weeks)	ORR	85% (12 months)	71% (12 months)	55%	NA
CheckMate 142	II	Nivolumab + Ipilimumab	Treatment-naïve MSI-H mCRC (Nivolumab 3 mg/kg every 2 weeks + Ipilimumab 1 mg/kg every 6 weeks)	ORR	79% (24 months)	74% (24 months)	69%	NA
NICHE	II	Nivolumab + Ipilimumab	Resectable stage I–III MSI-H and MSS CRC	Safety Feasibility	NA	NA	Pathologic response rate MSI-H: 100% MSS: 27%	NA

**Table 2 cancers-14-02241-t002:** Non-exhaustive list of clinical trials including MSS mCRC patients and investigating PD-L1 expression together with ICIs-based treatment.

Clinical Trial	Immunotherapy	Target	Other Therapy	Biomarkers	Clinical Indication
NCT03927898	Toripalimab	PD-1	SBRT	PD-1, PD-L1, Ki-67, TCR-repertoire	mCRC
NCT01772004	Avelumab	PD-L1	NA	PD-L1	Adv. Solid tumors
NCT04432857	Pembrolizumab	PD-1	AN0025 (EP4 antagonist)	PD-L1	Adv. Solid tumors
NCT02888743	Durvalumab Tremelimumab	PD-L1 CTLA-4	RT (low dose)	PD-L1 T cells infiltration RNA-seq TMB Circulating immune cells populations	mCRC
NCT04713891	Atezolizumab	PD-L1	KF-0210 (PGE4 antagonist)	PD-L1 CD3+ CD8+	Adv. Solid tumors
NCT05064059	Favezelimab Pembrolizumab	LAG3 PD-1	NA	PD-L1	mCRC
NCT02947165	PDR001	PD-1	NIS793 (anti-TGF-β)	TILs PD-L1	Adv. Malignancies

**Table 3 cancers-14-02241-t003:** Non-exhaustive list of clinical trials including MSS mCRC patients and investigating genomic, epigenomic, and transcriptomic biomarkers together with ICIs-based treatment.

Clinical Trial	Immunotherapy	Target	Other Therapy	Biomarkers	Clinical Indication
NCT03436563	Bintrafusp Alfa	Anti-PD-1/TGF-β trap	NA	CMS4	mCRC MSS CMS4, MSI-H mCRC, MSI-H non-CRC
NCT03152565	Avelumab	PD-L1	Autologous dendritic cell vaccine	Dynamic CMS modification	MSS mCRC
NCT04695470	Sintilimab	PD-1	Fruquitinib (VEGFR inhibitor)	TMB-H (≥5 mut/Mb)	MSS mCRC
NCT03638297	BAT1306 or Pembrolizumab	PD-1	Aspirin/Celebrex (COX inhibitor)	TMB-H or MSI-H	mCRC
NCT02842125	Pembrolizumab Nivolumab	PD-1	Ad-p53 (adenovirus) Chemotherapy	TMB Immune cells PD-L1, PD-L2	mCRC
NCT02628067	Pembrolizumab	PD-1	NA	TMB	Adv. Solid tumors
NCT04866862	Camrezilumab	PD-1	Fruquitinib	TMB	Refractory MSS CRC
NCT03150706	Avelumab	PD-L1	NA	POLE/POLD1	mCRC
NCT03435107	Durvalumab	PD-L1	NA	POLE/POLD1	mCRC
NCT03810339	Toripalimab	PD-1	NA	POLE/POLD1	Adv. Solid tumors
NCT03461952	Nivolumab Ipilimumab	PD-1 CTLA-4	NA	POLE/POLD1	Adv. Solid tumors with POLE/POLD1 mutations
NCT03767075	Atezolizumab	PD-L1	NA	POLE/POLD1	Adv. Solid tumors with POLE/POLD1 mutations
NCT03832621	Nivolumab Ipilimumab	PD-1 CTLA-4	Temozolomide	MGMT methylation TMB	MSS MGMT silenced mCRC
NCT03519412	Pembrolizumab	PD-1	Temozolomide	TMB	MSS (TMB ≥ 20 mut/Mb) or MSI-H mCRC
NCT04457284	Nivolumab	PD-1	Temozolomide Cisplatin	NA	MSS CRC

**Table 4 cancers-14-02241-t004:** Non-exhaustive list of clinical trials including MSS mCRC patients and investigating Immunoscore and immune infiltration as prognostic biomarker or predictive biomarker together with ICIs-based treatment.

Clinical Trial	Immunotherapy	Target	Other Therapy	Biomarker	Clinical Indication
NCT04938986	NA	NA	SOC	Immunoscore	Non-metastatic CRC
NCT01688232	NA	NA	SOC	Immunoscore	CRC
NCT03422601	NA	NA	Oxaliplatin	Immunoscore	Stage III
NCT02274753	NA	NA	SOC	Immunoscore NGS miRNA	CRC
NCT04262687	Pembrolizumab	PD-1	Bevacizumab Oxaliplatin	Immunoscore High immune infiltrate	MSS mCRC
NCT02646748	Pembrolizumab	PD-1	Itacitinib	TILs (CD8+, FOXP3+)	Adv. Solid tumors
NCT02512172	Pembrolizumab	PD-1	Azacitidine Romidepsin	TILs (CD4+ CD8+)	Adv. Solid tumors
NCT02837263	Pembrolizumab	PD-1	SBRT	TILs	Liver metastatic mCRC

**Table 5 cancers-14-02241-t005:** Non-exhaustive list of clinical trials investigating liquid biomarkers together with ICIs-based treatment.

Clinical Trial	Immunotherapy	Target	Other Therapy	Biomarkers	Clinical Indication
NCT03946917	JS001	PD-1	Regorafenib (Kinase inhibitor)	ctDNA	Adv. CRC
NCT04046445	BI754091	PD-1	ATP128 (Vaccine)	ctDNA	MSS mCRC
NCT02997228	Atezolizumab	PD-L1	Bevacizumab Chemotherapy	ctDNA Dynamic TCR repertoire PD-L1 MLH1	MSI-H mCRC
NCT03927898	Toripalimab	PD-1	SBRT	Dynamic TCR repertoire PD-L1 tumor cells PD-1, Ki-67 T cells	mCRC
NCT04714983	DNX-2440	OX40		T cells infiltration Dynamic TCR repertoire	mCRC
NCT02713373	Pembrolizumab	PD-1	Cetuximab	T cells populations (Flow cytometry)	mCRC
NCT03984578	Pembrolizumab	PD-1	Chemotherapy	T cells populations (Flow cytometry)	CRC
NCT02851004	Pembrolizumab	PD-1	Napabucasin (STAT3 inhibitor)	T cells populations (Flow cytometry) CMS	MSS/MSI mCRC
NCT05086692	ICI	NA	MDNA11 (IL-2 superkine)	T cells populations (Flow cytometry)	Adv. Solid tumors
NCT04348643	CAR T cells	CEA	NA	T cells populations (Flow cytometry)	CEA+ CRC
NCT02349724	CAR T cells	CEA	NA	CAR T cells (Flow cytometry)	CEA+ CRC
NCT04513431	CAR T cells	CEA	NA	CAR T cells (Flow cytometry)	CEA+ CRC
NCT03638206	CAR T cells	c-MET	NA	CAR T cells (Flow cytometry	CRC

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
