# Peer review of "Biomarkers of Response and Resistance to Immunotherapy in Microsatellite Stable Colorectal Cancer: Toward a New Personalized Medicine"

_cancers, 2022, doi:10.3390/cancers14092241_

Round 1

Reviewer 1 Report

This manuscript reviews the current state of play for immune checkpoint inhibitor use in microsatellite stable colorectal cancer.

The review is thorough and interesting to read. I have only minor points to raise.

  1. There are minor typographical and grammatical errors that should be corrected at proofing.
  2. The authors should ensure all acronyms are defined at first use.
  3. line 375-376 - comment about PD-L1 antibodies and conditions is 'tacked on' to the rest of this sentence. Could be separated into its own sentence or 2 and additional discussion on cut-offs and scoring of tumour cells alone vs tumour+immune cells. The latter point is raised later but also relevant here regarding non-standardised scoring.
  4. Some of the challenges of liquid biopsy have not been addressed - for TMB it can be difficult to distinguish between a true increase in TMB vs increase ctDNA burden in the bloodstream. This could be discussed.
  5. The authors appear to raise both tumour intrinsic microbiome and broad gut microbiome. The second is poorly addressed and in section 4.7 the writing shifts between the two without adequate context. There should be better explanation of the difference specifically for CRC given the close interaction of the 'gut microbiome' with the tumour site and work on broad microbiome deserves more discussion.

Reviewer 2 Report

In present review, authors discusses on the influence of certain cofounders on efficacy of immunotherapy in microsatellite stable colorectal cancer patients. I have several reservations. My comments are appended as below:

  1. Line 39, reference 3- annotate the quantitative (survival time, statistical inference as HR, P value if available) data on survival.
  2. Line 41- annotate with reference.
  3. Immunotherapy response in microsatellite stable vs unstable cancers- how is the difference in prognosis? Are there other cofounders as stage/age of patients noted? Where ever possible, the prognosis should be quantitatively described. For instance, line 80, reference 14.
  4. In connection with above comment, additional cofounders such as diet, smoking, body weight are known to affect the immunotherapy efficacy. Are these factors known to affect microsatellite stable vs unstable colorectal cancer patients prognosis on immunotherapy? Authors may refer PMID: 33076303 and discuss.
  5. Genomic and epigenomic classifications: authors should attempt to add figure describe the mechanistic part, it may have ease in understanding.
  6. Line 236- specify the miRNA and associated signaling.
    7. Tables 1-4- authors should add statistical inference (HR, P value) if available.
  7. Line 251-262- annotate with reference.
  8. Tumor infiltrating lymphocytes and Immunoscore: discuss on the Th17, and memory T cells.
  9. section 4.8.4. Cytokines- specify the source of cells secreting these cytokines. In this connection role of fibroblasts should be emphasized.

11.There should be future directions section.

Round 2

Reviewer 2 Report

All my concerns are addressed.